# Spectrophotometric Investigations of Charge Transfer Complexes of Tyrosine Kinase Inhibitors with Iodine as a σ-Electron Acceptor: Application to Development of Universal High-Throughput Microwell Assay for Their Determination in Pharmaceutical Formulations

**DOI:** 10.3390/medicina59040775

**Published:** 2023-04-17

**Authors:** Ibrahim A. Darwish, Hany W. Darwish, Awadh M. Ali, Halah S. Almutairi

**Affiliations:** 1Department of Pharmaceutical Chemistry, College of Pharmacy, King Saud University, P.O. Box 2457, Riyadh 11451, Saudi Arabia; 2Department of Analytical Chemistry, Faculty of Pharmacy, Cairo University, Kasr El-Aini St., Cairo 11562, Egypt

**Keywords:** tyrosine kinase inhibitors, Iodine, charge-transfer complex, 96-Microwell spectrophotometric assay, high-throughput analysis

## Abstract

*Background and Objective*: Tyrosine kinase inhibitors (TKIs) are used for the treatment of different types of cancers. The current study describes, for the first time, the ultraviolet-visible spectrophotometric investigation of charge transfer complexes (CTCs) of seven TKIs, as electron donors, and iodine, as σ-electron. *Materials and Methods*: The formation of CTCs was promoted in dichloromethane, among the other solvents used in the investigation. The molar absorptivity values, association constants, and free energy changes of the CTCs were determined. Stoichiometric ratio of TKI: iodine as well as TKIs site(s) of interaction were addressed. Reaction was the basis for constructing a novel simple and accurate 96-microwell spectrophotometric assay (MW-SPA) with high-throughput property for the quantitative determination of TKIs in their pharmaceutical formulations. *Results*: Beer’s law, which relates CTC absorbances to TKI concentrations, was followed within the optimal range of 2 to 100 µg/well (r ranged from 0.9991 to 0.9998). Detection and quantification limits ranged from 0.91 to 3.60 and 2.76 to 10.92 g µmL^−1^, respectively. Relative standard deviations values for the intra- and inter-assay precisions of the proposed MW-SPA did not exceed 2.13 and 2.34%, respectively. Studies of recovery demonstrated MW-SPA accuracy, with results ranging from 98.9% to 102.4%. All TKIs, both in bulk form and in pharmaceutical formulations (tablets), were effectively determined using the suggested MW-SPA. *Conclusions*: The current MW-SPA involved a simple procedure and it was convenient as it could analyse all proposed TKIs utilizing a single assay system at once measuring wavelengths for all TKIs. In addition, the proposed MW-SPA has high throughput which enables the processing of a batch of huge samples’ number in very short reasonable time period. In conclusion, TKIs can be routinely analysed in their dosage forms in quality control laboratories, and the assay can be highly valuable and helpful in this regard.

## 1. Introduction

Cancer is the world’s second cause of death for both men and women. In 2022, it was reason standing behind ~10 million fatalities (13% of all deaths). Cancer fatalities are expected to rise further, by 2030 [1]. Cancer has become an ever-increasing concern, according to studies [2,3,4], and it is internationally increasing, putting enormous burden on people, families, society, and health-care systems [5]. Most health systems in poor nations are ill-equipped to deal with this load, and some cancer patients throughout the world do not possess rights to immediate diagnosis and treatment; these countries account for over 70% of cancer fatalities [6]. Survival rates for many forms of cancer are increasing in nations with strong health systems, due to early identification, quality treatment, and survival care [1,6].

Surgery, radiation therapy, and chemotherapy are all potential components of cancer treatment. Isolated or localized cancers respond well to surgery and radiation, while more widespread tumours are treated with chemotherapy. All chemotherapeutic drugs are cytotoxic because they disrupt the production or activity of essential proteins, RNA, or DNA in living cells. Since the chemotherapeutics’ primary goal is cell death, they naturally generate devastating toxic side effects. An ideal chemotherapeutic, on the other hand, would only target diseased tissues and cells while avoiding healthy ones. Most chemotherapeutic drugs are extremely and, often, generally hazardous, notably to cells with short half-lives, thus idealistic application are not yet the reality [7]. Research in the life sciences’ fields has made great strides in recent years, shedding light on fundamental processes such as signal transduction in cancer cells, apoptotic activation, gene expression and more [8]. Furthermore, these medications can be very specific for particular cell targets (such as DNA or tubulin) that are present in both cancerous and healthy tissue. Researchers are concentrating on the primary enzymes of the signal transduction system which is crucial to the growth of malignant cells in order to develop more effective, safe, and specific innovative targeted drugs [9].

The tyrosine kinase protein family catalyses the phosphorylation of numerous essential proteins by transferring phosphate groups from ATP to tyrosine residues [10]. These phosphorylated proteins subsequently convey signals that control cell proliferation, differentiation, and death. Several disorders of the body have been linked to TK anomalies. More than half of oncogenes have been shown to exhibit TK activities, and their aberrant expression can lead to cellular proliferative problems and ultimately cancer [11]. Furthermore, tumour invasion, metastasis, and other symptoms are all linked to aberrant TK expression [12]. This is why scientists are currently focusing more on TK as a potential target for brand new chemotherapies. Institutions dedicated to drug discovery have placed a priority on tyrosine kinase (TK) as a target, with an emphasis on TK inhibitors (TKIs) that influence specific biochemical pathways [13]. The goal of developing TKIs was to block the activation of TK by competitively binding to the enzyme’s ATP-binding site. First approved in 2001, imatinib opened the door to new ways of thinking about cancer management [14,15,16]. The Food and Drug Administration (FDA) has now approved more than 20 different TKIs [17]. Some of these drugs are the primary prescription for treating cancer because they are very effective, have few adverse effects, and are highly targeted to certain types of cancer [10,11,12,13,14,15,16,17].

The quality of pharmaceutical formulations of TKIs in terms of potency and uniformity is crucial to the efficacy and safety of TKI therapy. An appropriate analytical method is needed for TKI quality control (QC). Chromatographic methods including high-performance liquid chromatography (HPLC) [18,19,20,21,22,23] and high-performance thin-layer chromatography (HPTLC) [24,25,26,27] as well as other techniques such as voltammetry [28], spectrofluorimetry [29,30,31,32], and spectrophotometry [33,34,35,36,37,38,39,40,41,42,43,44] have all been reported as methodologies for the QC of TKIs in pharmaceutical formulations. Because of its simplicity, low cost, and widespread availability in QC laboratories, spectrophotometry is the most practical analytical method. However, the majority of these assays rely on measuring native UV absorption, which has limited method selectivity [33,34,35,36,37,38,39]. There are just a handful of visible-spectrophotometric tests for TKIs, all of which involve derivatizing the compounds with a variety of chromogenic reagents [40,41]. Unfortunately, many tests necessitated lengthy extraction processes and large amounts of costly and hazardous chemical solvents [42,43,44,45], such as those relying on the creation of ion-pair associates [40]. Furthermore, because of variances in the chemical structures of the TKIs, these assays were independently designed. In addition, these tests adopted the classic spectrophotometric approach, that has constrained analytical throughputs. It would be practical and cost-effective to develop a novel and universal spectrophotometric assay for the determination of any TKI, regardless of its structure. Our work aims to address the limitations of formerly reported spectrophotometric methodologies for TKIs by establishing a novel, 96-microwell-based spectrophotometric assay (MW-SPA) that may be utilized in QC labs for the precise quantitation of any TKI. To create such an MW-SPA using charge-transfer complexes (CTCs) between iodine and TKIs, the existing technique makes use of iodine, a universal chromogenic reagent. The chemical structures of the seven TKIs used to design and validate the assay are shown in Figure 1. Table 1 provides their respective chemical names, molecular weights and molecular formulas.

## 2. Experimental

### 2.1. Apparatus

A double beam ultraviolet-visible spectrophotometer (UV-1800, Shimadzu Co., Ltd., Kyoto, Japan) with matched 1-cm quartz cells was utilized for scanning the UV-visible spectra. An absorbance microplate reader (ELx808: Bio-Tek Instruments Inc., Winooski, VT, USA) operated using KC Junior software was provided with the instrument.

### 2.2. Chemicals, Materials and Pharmaceutical Formulations

The chemicals, materials, and pharmaceutical formulations used in this study, along with their supplier/manufactures are given in Table 2. Lab-made CER and LIN tablets were prepared by a manual mixing of the active pharmaceutical substance and excipients; the quantity of TKI and those of the excipients are detailed in Table 2.

### 2.3. Preparation of TKIs Standard Solutions

Standard solutions (5 mg mL^−1^) of 6 cited TKIs namely AXT, CED, LIN, CAB, AFA and CER were prepared by dissolution of 50 mg of the standard material in ~ 1 mL methanol and completed to 10 mL with dichloromethane. A standard solution of PEL, because of its low solubility of PEL, was prepared at a concentration of 0.5 mg mL^−1^ by dissolving 5 mg of PEL in ~1 mL methanol and completed to 10 mL with dichloromethane. Stock solutions were stable for 14 days in storage in a refrigerator.

### 2.4. Preparation of TKI Tablets Solutions

An amount of 50 mg of tablet powder equivalent to all TKIs except PEL (5 mg) was relocated into a 10-mL calibrated flask, ~1 mL methanol and 5 mL of dichloromethane were added, and the mixture was shaken very well for 5 min. Volume was completed with to the mark with dichloromethane, shaken for 10 min, and filtered. The first part of the filtered solution was thrown away, and then a precise portion was diluted with dichloromethane to make TKI concentrations between 2 and 100 μg mL^−1^.

### 2.5. Determination of Association Constants

A series of solutions in the range of (0.22 × 10^−5^–6.17 × 10^−5^ M) were prepared in dichloromethane; the individual concentration range of each particular TKI is given in Table 2. Each one of these solutions was swirled with 1.58 × 10^−3^ M iodine solution. The reaction between TKIs and iodine was observed to be instantaneous at 25 ± 2 °C. Absorption spectra of these solutions were recorded against similarly treated reagent blank solutions. The absorbances measured at the maximum absorption peak (292 nm) were utilized to construct a Benesi-Hildebrand plot [46] of TKI-iodine CTC by putting [A]/A values versus l/[D]. The Benesi-Hildebrand equation [46] was utilized in order to carry out a linear regression analysis on the collected data. The equation is as follows:$$\frac{\left[A\right]}{A}=\frac{1}{\varepsilon }+\frac{1}{K\times \varepsilon }\times \frac{1}{\left[D\right]}$$
where [A]: iodine molar concentration; [D]: TKI molar concentration; A: absorbance of the CTC; ε: CTC molar absorptivity; K: CTC association constant (L mol^−1^). The intercept of the equation was =1/ε and association constants were computed from the values of both the equation slope and ε.

### 2.6. Molar Ratio

The reaction ratio was calculated using the spectrophotometric titration [47] technique. Both TKIs (2.5 × 10^−6^ M) and iodine (2 × 10^−5^ M) master solutions were made (i.e., molar concentration of iodine was 8-folds that of the TKI). Comparatively, the concentration of iodine solution was 8 × 10^−6^ M and that of PEL was 1 × 10^−6^ M. Master solutions of TKIs and iodine were prepared with a constant TKI concentration but with varying iodine concentrations, yielding a range of TKI: iodine molar ratios from 0.25 to 8 folds. Absorbances of CTC were recorded at 292 nm, and reactions were conducted at 25 ± 2 °C. An absorbance-versus-drug concentration curve was constructed. The molar ratio for the TKI/iodine reaction was determined using an established plot. When the tangents of the straight lines meet, that is the mole ratio.

### 2.7. Procedure of MW-SPA

Into the wells of the assay plates, 100 µL of iodine solution (0.1%, *w*/*v*) was dispensed, and then 100 µL of standard or tablet sample solution containing varying concentrations of TKI (2–100 µg mL^−1^) was added. The reader measured absorbances at 292 nm. Identical procedures were followed for the blank wells, but 100 µL of dichloromethane was dispensed instead of the sample, and the blank wells’ readings (A) were deducted from that of the sample wells.

## 3. Results and Discussion

### 3.1. Strategy and Design of the Study

Given their therapeutic value and the pressing need for a universal test for their estimating in the dosage forms, we chose TKIs as target analytes in our present study, despite their different chemical structures. Because of the relative ease of use and widespread adoption by QC laboratories, the creation of a spectrophotometric test for TKIs was suggested. Since all TKIs have many potential electron-donating sites in their chemical structures (Figure 1), which can form CTC with electron acceptors, our study considered the CT reaction of TKIs.

Iodine works as an electron acceptor and forms molecular complexes having a wide variety of compounds with electron-donating properties [48,49,50,51,52,53]. These studies demonstrated that iodine is a highly reactive σ-acceptor by virtue of its immediate reactions at ambient temperature. Investigations of the CT reaction of TKIs with iodine have not been investigated yet Therefore, iodine was chosen as an electron acceptor for the experiments in this study. The current study aimed to develop a new spectrophotometric assay for TKIs, free of limitations of traditional CT-based assays, such as low throughput and the consumption of significant quantities of organic solvents, which are both costly and harmful to analysts [42,43,44,45]. This objective was accomplished by varying out CT reaction between TKIs and iodine on 96-well assay plates and monitoring absorbances using an absorbance microplate reader. With this method, analysts may quickly run several samples and acquire a huge amount of data, which is ideal for QC labs but can be impactful on time, energy, and reagents if manually carried out [54,55].

### 3.2. UV-Visible Absorption Spectra

TKI solutions are colorless, having UV-visible absorption spectra between 200 and 800 nm. The spectra were distinctive in their overall form, maxima of absorption, and molar absorptivities. There was a noticeable difference in their absorption spectra because of the unique chemical structures of each (Figure 1). None of the TKIs we investigated had UV-absorption ≥ 400 nm, despite having widely varied absorption spectra. As an illustration, we used the CAB spectrum (Figure 2). The absorption spectra of a violet iodine solution in dichloromethane showed two peaks (λ_max_) at 231 and 505 nm. When TKI solutions were combined with iodine, this violet color quickly vanished and was replaced by a bright lemon yellow. Absorption peaks were observed at 292 and 359 nm in the reaction mixtures’ spectra, with the peak at 292 nm being 1.75 times higher than the peak at 359 nm (Figure 2). Since the I^3−^ absorption spectrum in dichloroethane exhibited same absorption maxima, researchers concluded that the fading of the iodine’s violet color and generation of these two new peaks in the absorption spectra of the reaction solutions were due to CTC the generation between TKIs and iodine, with ionized structure DI^+^......I^3^ (D represents electron donor; TKI). It was determined that this complex would originated from the outer complex D....I_2_.



$\begin{array}{lllllll} {D+I}_{2} & ⇌ & {\;D-I}^{+}I^{-} & ⇌ & {\;[D-I}^{+}]{+I}^{-} & ⇌ & {\;I}_{3}^{-}\\ & & \;\mathrm{Outer} & & \;\mathrm{Inner} & & \;\mathrm{Tri}-\mathrm{iodide}\\ & & \;\mathrm{complex} & & \;\mathrm{complex} & & \;\mathrm{ion}\;\mathrm{pair} \end{array}$



Because of the high intensity at 292 nm more than at 359 nm, all consequent measuerments and calculations were carried out at 292 nm.

### 3.3. Optimizing the Solvent of the Reaction

Absorption spectra were captured during CT reactions of TKI with iodine and the production of the tri-iodide ion pair in a variety of non-polar solvents, including carbon tetrachloride, cyclohexane, dichloroethane, dichloromethane, and chloroform. In each solvent, we calculated the molar absorptivity (ε) and λ_max_. As expected, the results demonstrated some variation in λ_max_ and ε values of. Dichloromethane yielded the greatest increases in ε. Polar solvents such as methanol, ethanol, acetonitrile, etc., were not good for CTC formation because reagent blanks with iodine presented high absorbances. Following these results, all further experiments were carried out in dichloromethane.

### 3.4. Complex Parameters and Constants

We calculated the complex’s band gap energy, or Eg. “It is the least amount of energy needed to get an electron excited and move it from the lower energy valence band to the higher energy conduction band [56]”. Eg was determined by graphing the energy values (hυ, in eV) versus (αhυ)^2^ in Tauc plots derived from the absorption spectra of the TKI-iodine complexes (Figure 3). To determine Eg, we linearly extended the linear part in the chart to the point where (αhν)^2^ = 0 [57]. Among all TKIs, the obtained values of Eg fell between 3.62 and 3.76 eV. (Table 3). These numbers point to the simplicity of the electron transfer from TKI molecules to iodine and the subsequent development of the new absorption bands.

Using the Benesi-Hildebrand method [46], association constants (K_c_) were calculated at 25 ± 2 °C and at λ_max_ of TKI-iodine complexes (292 nm). Using the resulting straight lines (Figure 4), the K_c_ of CTC were computed. K_c_ values were between 2.23 × 10^2^ and 6.54 × 10^3^ L mol^−^^1^ (Table 3).

The standard free energy change (G_0_) of TKI-iodine complexes is proportional to (K_c_) and can be calculated by this formula: ΔG^0^ = −2.303 RT log K_c_

Where ΔG_0_: standard free energy change of complex (Kilo joules; KJ mole^−1^); R: gas constant (8.314 KJ mole^−1^); T: absolute temperature in Kelvin (°C + 273) and K_c_: association constant (L mole^−1^). ΔG^0^ values were comparable to all TKIs (~4.74 × 10^4^ J mole^−1^). These ΔG^0^ values revealed the easiness of interaction between TKIs and iodine and the stability of the TKI-iodine complex [58]. It was found that the values of ΔG^0^ inversely correlated with those of the K_c_ (ΔG^0^ decreases as K_c_ increases; Figure 5A), and the correlation was governed by a linear equation relationship correlating ΔG^0^ with ln K (Figure 5B).

### 3.5. Molar Ratio and Sites of Interaction

According to the reported spectrophotometric titration method [50], molar ratios of TKI to iodine were found to be 1:2 for AFA and CER and 1:1 for the other TKIs (Table 4). CED and AFA were used to illustrate the 1:1 and 1:2 molar ratios, respectively, and their determination from the spectrophotometric titration graph was described (Figure 6). Electron density on each atom was estimated using “CS Chem3D Ultra, version 16.0 (CambridgeSoft Corporation, Cambridge, MA, USA)” in conjunction with “molecular orbital computations software (MOPAC) and molecular dynamics computations software (MM2 and MMFF94)”. TKI’s atoms with the highest electron densities were designated as the most likely places to contribute to the interaction with iodine (Table 4). These data coincided with our previous study [59].

### 3.6. Optimization of MW-SPA Conditions

For maximizing the chance of a successful reaction in the 96-microwell assay plate, we performed several rounds of experiments in which we varied one reaction variable at a time while holding other variables constant. Dichloromethane was used for the reaction, and a plate reader set to 292 nm was utilized for all of the observations. After experimenting with different iodine concentrations and reaction times at 25 ± 2 °C, it was determined that a concentration of between 0.05 to 0.2% (*w*/*v*) was optimal (see Figure 7). Though similar investigations were directed to determine the optimum reaction time, it was discovered that the reaction instantly took place; nonetheless, measurements were taken 5 min after the reaction began to improve readability for the analyst. Table 5 provides a brief summary of the conditions evaluated and the optimal value chosen for the design of the suggested MW-SPA.

### 3.7. Validation of MW-SPA

#### 3.7.1. Linear Range and Sensitivity

The linear regression of data was carried out using the “least-squares method” under specified optimal parameters of MW-SPA (Table 5). The resulting calibration curves are shown in Figure 8. In the range of 2–100 μg mL^–1^, curves were linear with high correlation coefficients. Table 5 displays the linear parameters. International Council for Harmonization (ICH) recommendations [60] were adopted to establish the limits of detection (LOD) and quantitation (LOQ). The limits of detection and quantification (LOD and LOQ) ranged from 0.91 to 3.60 and 2.76 to 10.92 μg mL^–1^, respectively (Table 6).

#### 3.7.2. Precision and Accuracy

Samples of TKIs’ solutions were used to verify the method’s precision. Precision within (intra-) and between (inter-) assays had an RSD of 1.02–2.13 and 1.48–2.34%, respectively. These relatively low RSD values provided conclusive evidence of the assay’s high degree of precision. Recovery (R%) trials with constant amounts of TKIs were employed to assess assay accuracy. Table 7 shows that the R% ranged from 98.1 to 102.4 percent, illustrating the method’s accuracy.

#### 3.7.3. Robustness and Ruggedness

The test’s robustness was assessed, which is defined as the degree to which tiny changes in the assay variables have no tangible effect on the assay’s results. Changes of 10% were made to the optimal values of three variables: iodine concentration, reaction time, and temperature (Table 5). Results of the assay were found to be insensitive to variations in the technique variables as recoveries were nearly 100%. The results validated the proposed assay’s suitability for routine application in the analysis of TKIs.

Two separate analysts performed the runs over the course of three days to ensure ruggedness. The greatest RSD values from day-to-day changes did not surpass 3%, indicating that the results were reproducible.

#### 3.7.4. Specificity and Interference

The advantage of the proposed MW-SPA is that readings in the visible region can be accomplished independently of UV-absorbing excipients that might be co-extracted from pharmaceutical preparations containing TKI. The probable interference of dosage form excipients was also investigated. Mixing known concentrations of TKI with varying amounts of common excipients to prepare analytical samples. These included common excipients such as “microcrystalline cellulose, colloidal silicon dioxide, anhydrous dibasic calcium phosphate, sodium starch glyconate and magnesium stearate”. The R% ranged from 97.5 to 103.5 percent, as shown in Table 8, indicating that none of the excipients exhibited any interaction with the indicated technique. The absence of interference from these excipients resulted from the extraction of TKI from samples using an organic solvent in which these excipients do not dissolve.

### 3.8. Application of MW-SPA in the Analysis of TKIs in Pharmaceutical Formulations

The obtained validated results proved to be sufficient to make the proposed MW-SPA appropriate for routine analysis of TKIs in QC labs. Table 8 displays the results of the proposed MW-SPA to determine TKIs in a variety of pharmaceutical formulations. The average values for labeled quantities reported were 99.52.1% to 1021.8% (Table 9). These findings supported the assumption that the suggested MW-SPA could be utilized to accurately assess TKIs in tablet form.

## 4. Conclusions

The charge transfer reaction of TKIs with iodine was investigated by UV-visible spectrophotometry. The results revealed the formation of CTCs, as evidenced by the appearance of new absorption bands in the absorption spectra of the reaction mixtures. The molar ratios of the CTCs (TKI:iodine) were 1:2 for AFA and CER, and it was 1:1 for the other TKIs. The high association constants with low free energy changes proved the ease of formation of the CTCs and their stability. The reaction between TKIs and iodine was adapted as a basis for the novel MW-SPA for the accurate and precise determination of TKIs in in their dosage forms (Tablets). The proposed MW-SPA is applicable to all TKIs despite the variation in their chemical structure without adjustments to the detection wavelength. The proposed assay has many benefits, including its ease of use, the low cost of the analytical reagents it employs, the low amounts of reagent and solvent it requires, and its low toxicity. It is wise mention that the assay described herein this study has a higher sensitivity, but narrower range, than the previous assays employing chloranilic acid as an electron acceptor [59]. The linear range of the present and previous assays were 2–100 and 5–500 µg/well, respectively. Furthermore, the LOQ values were 0.91–10.92 and 5.82 and 15.42 µg/well, respectively.

## Figures and Tables

**Figure 1 medicina-59-00775-f001:**
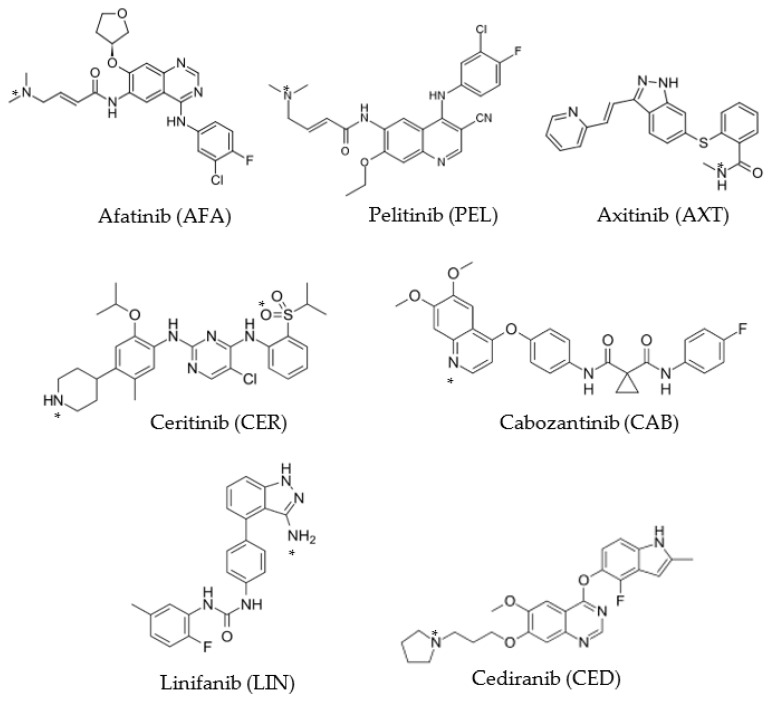
The chemical structures of the investigated tyrosine kinase inhibitors (TKIs) with their abbreviations. Asterisk symbols (*) denote the electron-donating sites of interactions of TKIs with iodine and form CTC.

**Figure 2 medicina-59-00775-f002:**
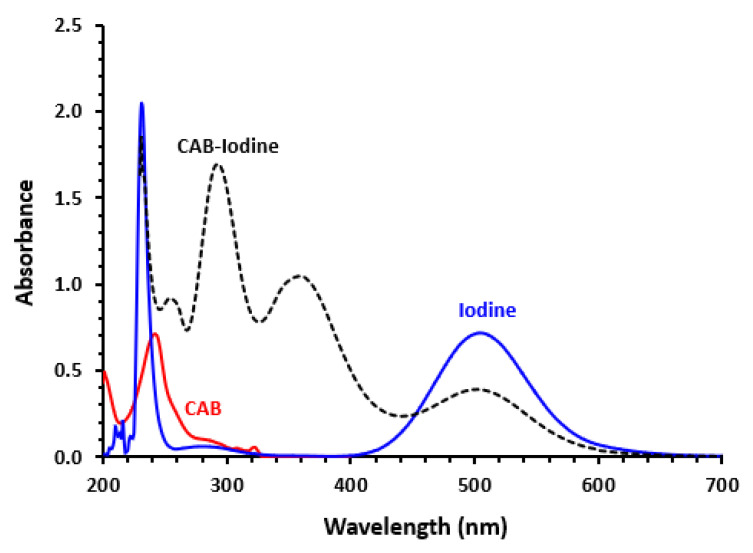
The absorption spectra of CAB (1.99 × 10^−5^ M), iodine (1.58 × 10^−3^ M), and a reaction mixture of CAB (3.99 × 10^−5^ M) with iodine (1.58 × 10^−3^ M). All solutions were in dichloromethane.

**Figure 3 medicina-59-00775-f003:**
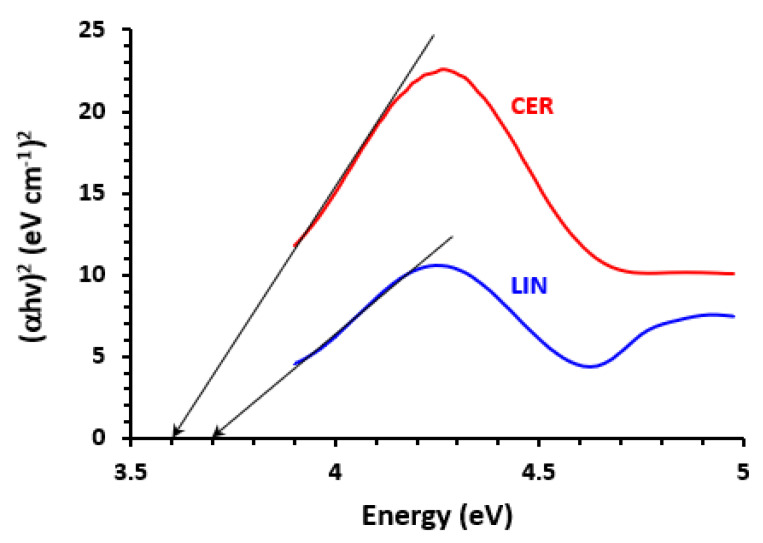
Tauc plot of energy (hυ, in eV) against (αhυ)^2^ against for CT complex of iodine with CER and LIN.

**Figure 4 medicina-59-00775-f004:**
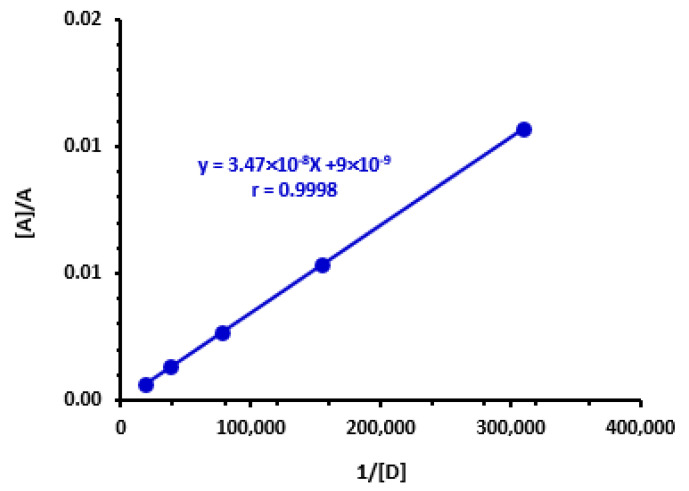
Benesi-Hildebrand plot of the CT complex of iodine with AXT and the linear fitting equation with correlation coefficient (r). [A], A and [D] are the molar concentration of iodine, the absorbance of the reaction mixture of the complex, and the molar concentration of AXT, respectively.

**Figure 5 medicina-59-00775-f005:**
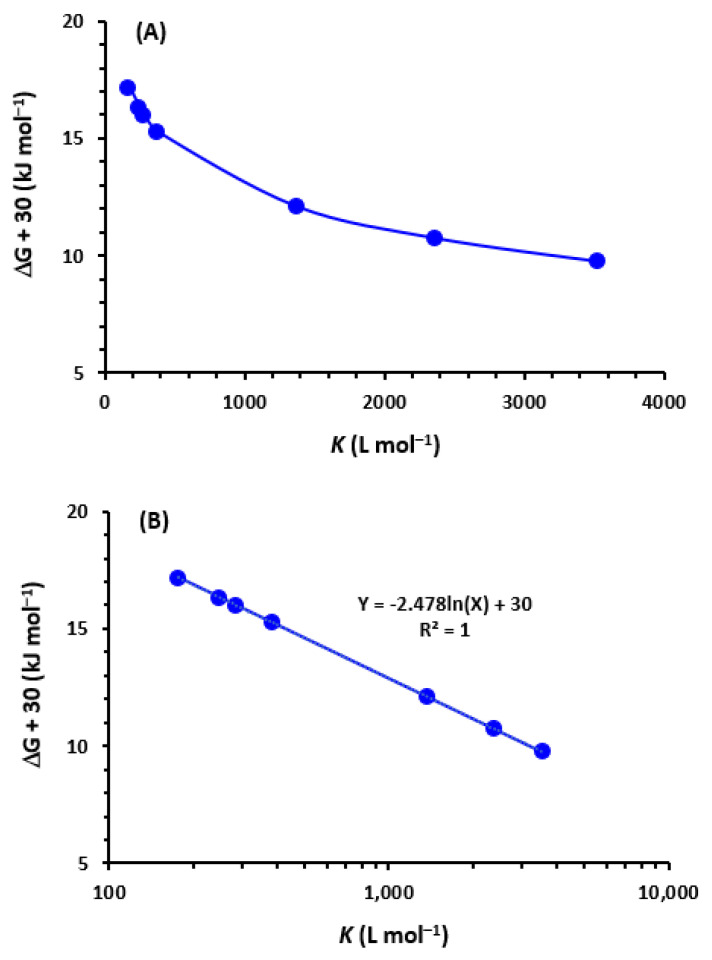
The association constant (*K*) versus free energy change (ΔG) for the CTC of TKIs with iodine. *K* values were presented in linear and logarithmic scale (panels (**A**) and (**B**), respectively). A value of 30 was added to all ΔG values to convert their negative values to positive ones.

**Figure 6 medicina-59-00775-f006:**
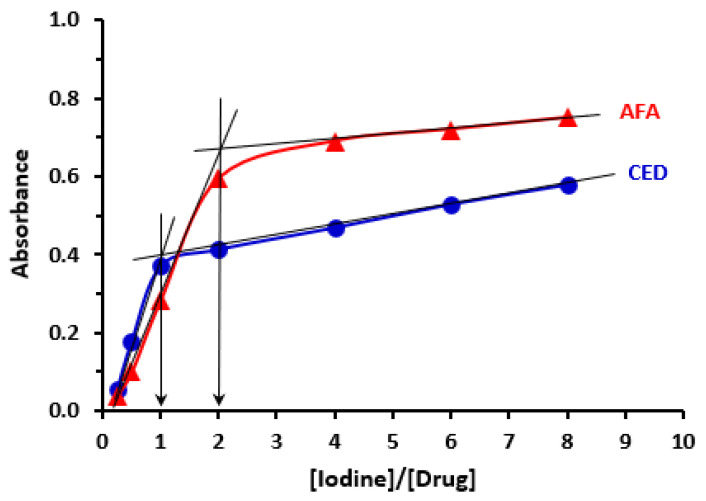
Plot of absorbance versus molar ratio of [Iodine]/[Drug] obtained from reaction mixtures containing a fixed concentration of drug and varying concentrations of iodine. The mole ratio corresponds to the point of intersection of the tangents of the straight-line portions of the plots. Results are given for CER and AFA as representative examples for the molar ratios of 1:1 and 1:2, respectively.

**Figure 7 medicina-59-00775-f007:**
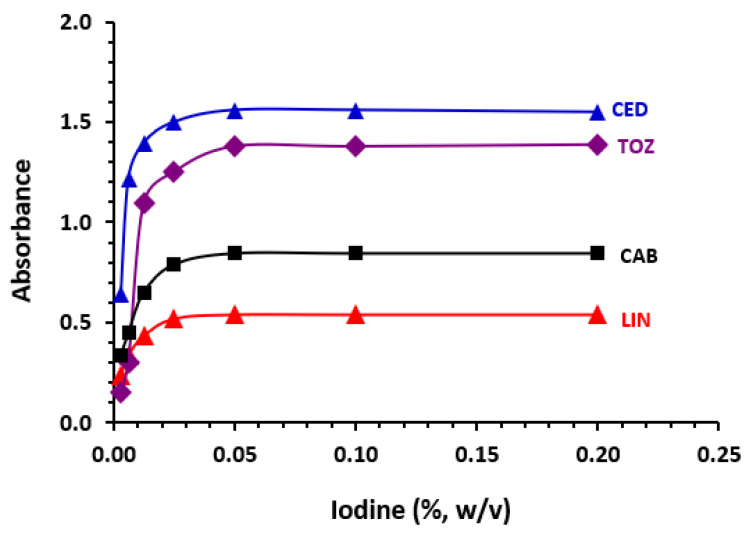
The effect of iodine concentration on its reaction with TKIs (25 µg mL^−1^).

**Figure 8 medicina-59-00775-f008:**
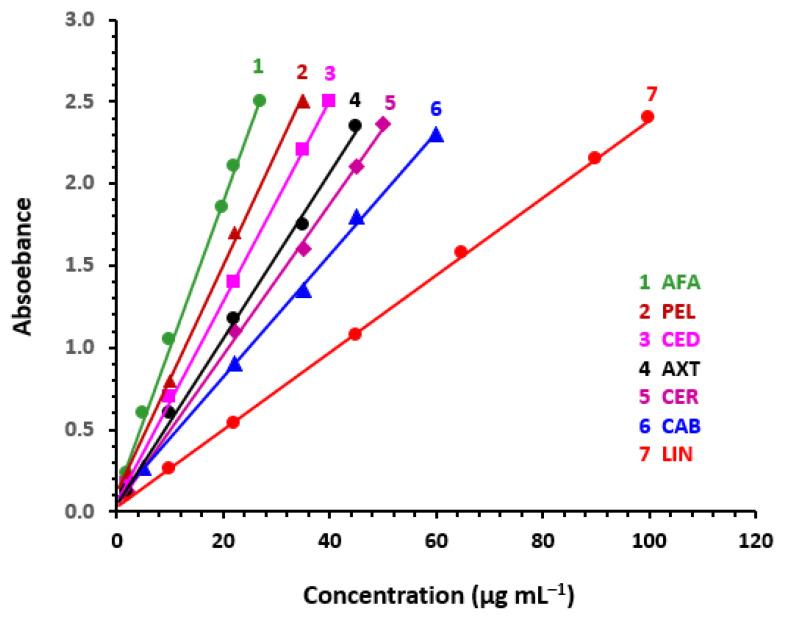
Calibration curves for the determination of TKIs by the proposed MW-SPA based on their reaction with iodine.

**Table 1 medicina-59-00775-t001:** The investigated TKIs with their abbreviations, (International Union of Pure & Applied Chemistry (IUPAC) names, molecular formulae and molecular weights.

TKI Name	Abbreviation	IUPAC Name	Molecular Formula	Molecular Weight
Afatinib	AFA	“(*Z*)-but-2-enedioic acid;(*E*)-*N*-[4-(3-chloro-4-fluoroanilino)-7-[(3*S*)- oxolan-3-yl]oxyquinazolin-6-yl]-4-(dimethylamino)but-2-enamide”	C_24_H_25_ClFN_5_O_3_	485.94
Pelitinib	PEL	“(2E)-*N*-{4-[(3-chloro-4-fluorophenyl)amino]-3-cyano-7- ethoxyquinolin-6-yl}-4-(dimethylamino)but-2-enamide”	C_24_H_23_ClFN_5_O_2_	467.90
Cediranib	CED	“4-[(4-fluoro-2-methyl-1H-indol-5-yl)oxy]-6-methoxy-7-(3-pyrrolidin-1-ylpropoxy)quinazoline”	C_25_H_27_FN_4_O_3_	450.51
Axitinib	AXT	*“N*-methyl-2-[[3-[(*E*)-2-pyridin-2-ylethenyl]-1*H*-indazol-6-yl]sulfanyl] benzamide”	C_22_H_18_N_4_OS	386.47
Ceritinib	CER	“5-chloro-2-*N*-(5-methyl-4-piperidin-4-yl-2-propan-2-yloxyphenyl)-4-*N*-(2-propan-2-ylsulfonylphenyl)pyrimidine-2,4-diamine”	C_28_H_36_ClN_5_O_3_S	558.14
Cabozantinib	CAB	“N′1-{4-[(6,7-dimethoxyquinolin-4-yl)oxy]phenyl}-N1-(4-fluorophenyl) cyclopropane-1,1-dicarboxamide”	C_28_H_24_FN_3_O_5_	501.50
Linifanib	LIN	“1-[4-(3-amino-1*H*-indazol-4-yl)phenyl]-3-(2-fluoro-5-methylphenyl)urea”	C_17_H_15_FN_5_O	324.34

**Table 2 medicina-59-00775-t002:** Chemicals, materials and pharmaceutical formulations used in the study.

Chemicals, Materials and Formulations	Manufacturer (Address)
TKIs (purities were > 99%)	LC Laboratories (Woburn, MA, USA)
Iodine	Sigma-Aldrich Chemicals Co. (St. Louis, MO, USA)
Transparent polystyrene 96-microwell plates	Corning/Costar Inc. (Cambridge, MA, USA)
Adjustable 8-channel-pipettes	Sigma-Aldrich Chemicals Co. (St. Louis, MI, USA).
Inlyta film-coated tablets (50 mg AXT/tablet)	Pfizer (New York, NY, USA)
Recentin tablets (30 mg CED/tablet)	AstraZeneca (Cambridge, UK)
Gilotrif tablets (40 mg AFA/tablet)	Boehringer Ingelheim (Ingelheim am Rhein, Germany)
Cabometyx tablets (40 mg CAB/tablet)	Exelixis, Inc. (Alameda, CA, USA)
Lab-made CER tablets (50 mg/table)	Prepared in the laboratory by mixing 50 mg of CER with 25 mg of starch, hydroxypropyl cellulose, microcrystalline cellulose and lactose monohydrate.
Lab-made LIN tablets (25 mg/tablet)	Prepared in the laboratory by mixing 25 mg of LIN with the same excipients used in preparing CER tablets.

**Table 3 medicina-59-00775-t003:** Spectrophotometric parameters of the CT reaction of iodine with TKIs.

TKIs	Concentration Range (M × 10^−5^)	Molar Absorptivity (ε_max_, (L mol^−1^ cm^−1^)	Band Gap Energy (eV)	Association Constant (*K*, L mol^−1^ × 10^2^)	Free Energy Change (ΔG^0^, kJ mol^−1^)
AFA	0.26–4.12	1.2 × 10^4^	3.74	4.96 × 10^3^	−21.09
PEL	0.27–4.27	8.75 × 10^3^	3.76	6.54 × 10^3^	−21.77
CED	0.22–3.58	1.17 × 10^5^	3.74	2.88 × 10^2^	−14.04
AXT	0.32–5.18	1.17 × 10^5^	3.74	2.47 × 10^2^	−13.66
CER	0.22–3.58	1.87 × 10^4^	3.62	2.23 × 10^2^	−19.10
CAB	0.25–3.99	7.97 × 10^3^	3.68	3.52 × 10^3^	−20.24
LIN	0.39–6.17	4.87 × 10^3^	3.72	2.37 × 10^3^	−19.26

**Table 4 medicina-59-00775-t004:** The molar ratios of the reaction of TKIs with iodine, types of atoms proposed site(s) of interaction on TKIs molecules and charges on these atoms.

TKI	Molar Ratio (TKI:Iodine)	Atom Type(s) Proposed as Site(s) of Interaction ^a^	Charge ^b^
AFA	1:2	(N31): Nitrogen of the tertiary dimethyl amine moiety	−0.81
		N(4): Nitrogen atom of aromatic of quinazoline ring	−0.62
PEL	1:1	(N24): Nitrogen atom of amine of dimethylamine moiety	−0.81
CED	1:1	(N25): Amine nitrogen of pyrrolidinyl ring	−0.81
AXT	1:1	(N27): Nitrogen atom of amide, N-C=O	−0.73
CER	1:2	(N2): Nitrogen atom of piprazine ring	−0.62
		(O36): one of oxygen terminals on sulfur atom	−0.65
CAB	1:1	(N3): Aromatic pyridine nitrogen of quinoline ring	−0.62
LIN	1:1	(N26): Nitrogen atom of primary amine atom, N-C=N	−0.88

^a^ These site(s) of interactions are denoted on the chemical structures of the TKIs (Figure 1). ^b^ The negative sign indicate the negative electron density.

**Table 5 medicina-59-00775-t005:** Optimization of experimental conditions for the MW-SPA for TKIs based on their CT reaction with iodine.

Condition	Studied Range	Optimum Value ^a^
Iodine conc. (%, *w*/*v*)	0.05–0.2	0.1
Solvent	Different ^b^	Dichloromethane
Reaction time (min)	0–40	5
Temperature (°C)	25–50	25
Measuring wavelength (nm)	200–700	292

^a^ Optimum values were used for all TKIs. ^b^ Solvents used were carbon tetrachloride, cyclohexane, dichloroethane, dichloromethane and chloroform.

**Table 6 medicina-59-00775-t006:** Calibration parameters for the analysis of TKIs by MW-SPA based on their CT reaction with iodine.

TKIs	Linear Range ^a^	Intercept	SDa ^b^	Slope	SDb ^b^	r ^b^	LOD ^a^	LOQ ^a^
AFA	2–30	0.1146	0.0245	0.0899	0.0141	0.9992	0.91	2.76
PEL	2–45	0.1192	0.0225	0.0690	0.0221	0.9994	1.08	3.27
CED	2–40	0.0653	0.0196	0.0610	0.0024	0.9998	1.06	3.21
AXT	2–45	0.0501	0.0219	0.0503	0.0054	0.9996	1.44	4.37
CER	2–50	0.0477	0.0181	0.0455	0.0042	0.9997	1.31	3.97
CAB	5–60	0.0706	0.0310	0.0377	0.0042	0.9991	2.71	8.22
LIN	10–100	0.0401	0.0249	0.0288	0.0031	0.9995	3.60	10.92

^a^ Values are in µg mL^–1^. ^b^ SDa = standard deviation of the intercept, SDb = standard deviation of the slope, r = correlation coefficient.

**Table 7 medicina-59-00775-t007:** Precision and accuracy of the proposed MW-SPA for TKIs via their CT reactions with iodine.

TKIs	Relative Standard Deviation (%) ^a^		Recovery (% ± SD) ^a^
	Intra−Assay, *n* = 3	Inter−Assay, *n* = 3	
AFA	1.54	1.65	101.4 ± 1.5
PEL	1.25	2.25	99.2 ± 1.2
CED	1.82	1.92	100.4 ± 1.6
AXT	1.04	1.54	99.8 ± 1.4
CER	1.02	1.48	98.9 ± 1.8
CAB	2.13	2.34	101.2 ± 1.5
LIN	1.46	1.62	102.4 ± 2.2

^a^ Values are mean of three determinations.

**Table 8 medicina-59-00775-t008:** Analysis of TKIs in the presence of the excipients are those present in their solid pharmaceutical tablets by the proposed MW-SPA based on their CT reactions with iodine.

Excipient ^b^	Recovery (% ± SD) ^a^			
	PEL	CER	CAB	LIN
MCC (50) ^c^	101.2 ± 1.5	100.2 ± 1.8	102.4 ± 1.1	103.6 ± 1.7
CSD (10)	98.8 ± 1.8	101.4 ± 1.9	100.2 ± 1.4	99.4 ± 1.4
ADCP (5)	102.5 ± 1.9	102.5 ± 2.4	99.6 ± 0.8	102.1 ± 1.5
SSG (5)	100.8 ± 1.2	99.4 ± 1.9	97.5 ± 2.4	103.2 ± 4.9
MS (5)	99.6 ± 2.4	103.5 ± 2.3	103.2 ± 1.2	99.3 ± 1.6

^a^ Values are the means of three determinations. ^b^ Abbreviations are: MCC = microcrystalline cellulose, CSD = colloidal silicone dioxide, ADCP = anhydrous dibasic calcium phosphate, MS = Magnesium stearate. ^c^ Figures in parenthesis are the amounts in mg added per 50 mg of TKI.

**Table 9 medicina-59-00775-t009:** Determination of TKIs in their pharmaceutical formulations by the proposed MW-SPA based on their CT reaction with iodine.

Labe Claim (% ± RSD) ^a^			
Gioltrif Tablets (40 mg AFA)	Recentin Tablets (30 mg CED)	Cabometyx Tablets (40 mg CAB)	Inlyta Tablets (50 mg AXT)
101.2 ± 1.6	100.5 ± 1.5	99.8 ± 1.4	102.1 ± 2.4
100.4 ± 1.8	101.2 ± 2.1	101.2 ± 2.2	99.6 ± 1.22
99.5 ± 2.1	99.6 ± 1.8	102.1 ± 1.8	100.8 ± 2.4

^a^ Values are mean of three determinations.

## Data Availability

All data are available in the article.

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
