# Peer review of "Spectrophotometric Investigations of Charge Transfer Complexes of Tyrosine Kinase Inhibitors with Iodine as a σ-Electron Acceptor: Application to Development of Universal High-Throughput Microwell Assay for Their Determination in Pharmaceutical Formulations"

_medicina, 2023, doi:10.3390/medicina59040775_

Round 1

Reviewer 1 Report

The authors has explained all the facts and principle well. The present manuscript can be accepted in the present form. However some spelling mistakes are there hence, it should be throughly checked for spelling and other english mistakes. 

Author Response

Please, find the attached file

Reviewer 2 Report

Line 25: r or r2??

The abstract can be trimmed

Just Iodine cant be consider under keywords

Lie 41- cancer statistics can be updated- it mentioned till 2018 only

Line 84- abbreviate HPLC and HPTLC

160- Write in equation format

Figure 4- not mentioned in the manuscript

The least information in Table 9 mentioned tablets-whether it is marketed or prepared-

if marketed mention the product details with the batch number

if prepared mention the formula

Author Response

Please, find the attached file

Reviewer 3 Report

Manuscript id medicina-2248682 authored by dr Darwish et al is an interesting research aiming to develop and validate spectrophotometric methods for quantification of several tyrosine kinase inhibitors (TKIs) in pharmaceutical dosage forms. Charge transfer complexes of afatinib, pelitinib, cediranib, axitinib, ceritinib, cabozantinib and linifanib with Iodine were studied. Manuscript seems well written and well documented. Let mi highlight some aspects that may increase the quality of your manuscript.

1. Please describe in detail the manufacturing of lab-made tablets 

2. I was able to trace an article published by your research team (https://doi.org/10.1016/j.saa.2021.119482). Table 4 in current manuscript is quite the same as in previously published article. Please made proper citation of your previously published paper. Also please compare analytical methods validation parameters (for the two methods based on chloranilic acid and iodine).

Author Response

Please, find the attached file

Round 2

Reviewer 3 Report

Dear authors,

You addressed all my major concerns. However I must insist on observation 1 from previous review report. The details about lab-made tablets and mentioned in Table 2 are not satisfactory:

"Lab-made CER tablets (50 mg/table)  - Prepared in the laboratory by mixing 50 mg of CER with 25 mg of starch, hydroxypropyl cellulose, microcrystalline cellulose and lactose monohydrate."

"Lab-made LIN tablets (25 mg/tablet)  - Prepared in the laboratory by mixing 25 mg of LIN with the same excipients used in preparing CER tablets."

Did you performed a manual mixing of API and excipients or the mixing of powders was done by means of a lab scale mixer? What tableting equipment was used for tablets preparation? Please detail. Or maybe you manufactured just the powder mix and you tested your method on the powder mix? Please revise your manuscript accordingly. 

Author Response

Please, find the attached file
